# Scattered Mixture-of-Experts Implementation

**Shawn Tan**
tanjings@mila.quebec

**Yikang Shen**
yikang.shen@ibm.com

**Rameswar Panda**
rpanda@ibm.com

**Aaron Courville**
courvila@iro.umontreal.ca

## Abstract

ScatterMoE is an implementation of Sparse Mixture-of-Experts (SMoE) on GPUs. ScatterMoE builds upon techniques in existing implementations, and overcoming some of the current limitations to improve batched inference, training speed, and memory footprint. This implementation achieves this by avoiding both padding and making excessive copies of the input. We also fuse expert linear transforms and reordering operations with `ParallelLinear`, a module that can be used to extend the concept of SMoEs. We benchmark our implementation against Megablocks, and show that it enables a higher throughput and lower memory footprint. We also show how `ParallelLinear` enables extensions of the Mixture-of-Experts concept via a demonstration with a Mixture of Attention implementation.

 https://github.com/shawntan/scattermoe

## 1 Introduction

Sparse Mixture of Experts (SMoEs; Shazeer et al. 2017) have become increasingly popular for scaling up Transformer-based language models. While applications of SMoEs like the Switch Transformer (Fedus et al., 2022) use SMoEs to scale "outrageously" large models by distributing computation of experts across compute nodes, it has proven useful even in scaling up smaller models where device memory is an issue.

For SMoEs, sparsity is key in reducing computation costs. However, fully exploiting the sparsity to improve the throughput of MoE modules is challenging. While a lot of deep learning research is implemented in PyTorch (Paszke et al., 2019), the naive implementation of SMoEs does not take full advantage of the parallelism of GPUs and are slow as a result. Initial implementations on Tensor Processing Units (TPUs) for Switch Transformers require all tensor sizes, or capacity, to be specified at compilation, which ensures that all load for every expert is equal (Fedus et al., 2022). This creates issues when experts are imbalanced: When the router assigns more tokens than capacity allows to a particular expert, some tokens are dropped. Likewise, when experts are underused, the tensors are padded, which creates unnecessary memory allocation. Later, Megablocks (Gale et al., 2023) and PIT (Zheng et al., 2023) framed the SMoE computation as a sparse matrix multiplication problem, which can be computed efficiently with sparse matrix optimised algorithms. In both these cases, the authors were able to create a more efficient GPU-based implementation of SMoEs.

Despite these recent advances, there is still room for improvement. First, existing implementations of SMoEs, performs a scatter-to-group initial copy of the input, creating a memory allocation overhead during training because of tensors stored for used in the backward pass. Some implementations pad the routed tensors so they are equal-sized blocks, which further increases the memory overhead. Second, Megablocks and PIT requires a translation of the SMoE problem into a sparse matrix format. While this incurs only a small part of the computation overhead, the sparse matrix format makes the intermediate representation harder to extend upon.

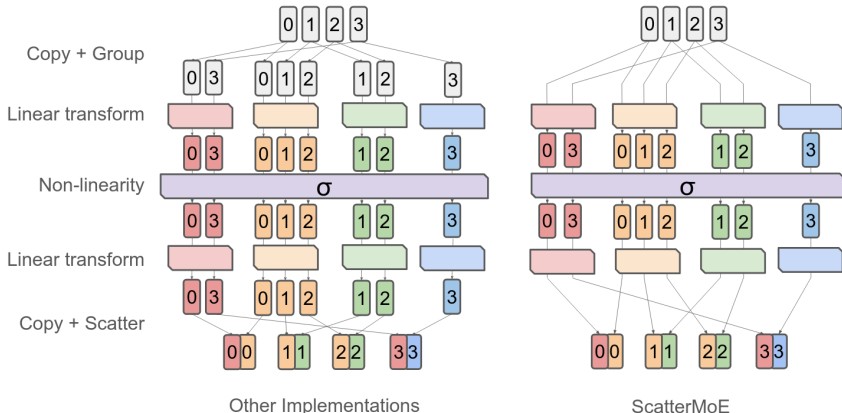

Figure 1: Current implementations of SMoE Multi-layer Perceptrons (MLPs) require a copy of the embeddings when grouping (left), while ScatterMoE fuses the grouping and linear transformation step (right), reducing the memory footprint of our method. The various colours represent different experts, while the vertical rectangular boxes represent embeddings with their associated time steps labelled above or below them.

In this paper, we present ScatterMoE, an SMoE implementation that minimises this memory overhead. This is made possible by `ParallelLinear`, a primitive we introduce that performs grouped matrix operations on scattered vectors. The resulting intermediate representations (*e.g.* hidden state of an SMoE MLP) can be exposed as standard PyTorch tensors, allowing for easy extensions of present SMoE methods to other types of expert modules. We demonstrate the utility of this representation by implementing SMoE Attention with `ParallelLinear`, following the specification in Tan et al. (2023). In the final section, we benchmark ScatterMoE against a naive PyTorch implementation and Megablocks.

## 2   Related Work

**Other implementations & Dependencies**   The core parts of ScatterMoE is implemented with Triton[1] (Tillet et al., 2019), a tile-based language for GPU programming in Python, making it the most accessible for modification and extension. Our main comparison is against Megablocks[2], which is implemented using the STK framework[3] which also uses Triton. Megablocks is also used in the Megatron-LM model (Shoeybi et al., 2019; Narayanan et al., 2021; Korthikanti et al., 2023), and its widespread use as an efficient method for training SMoEs. Another popular library for implementing SMoEs is CUTLASS[4](Kim et al., 2022), with which Megablocks uses as its `grouped` option.

---

[1] https://github.com/openai/triton
[2] https://github.com/stanford-futuredata/megablocks
[3] https://github.com/stanford-futuredata/stk
[4] https://github.com/NVIDIA/cutlass

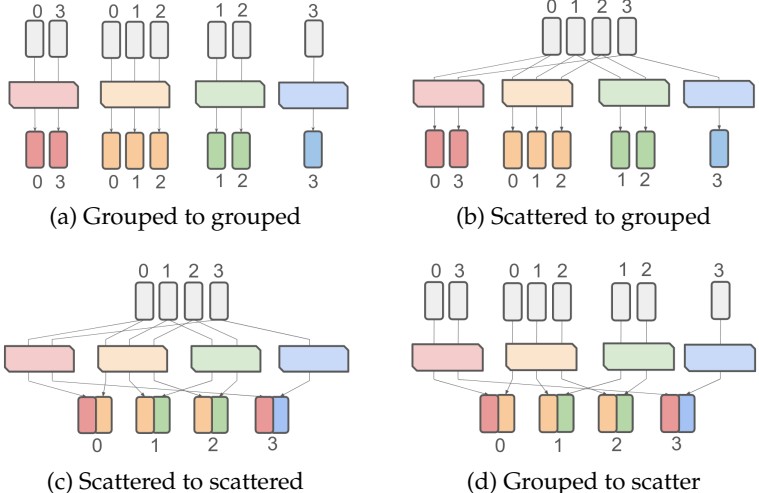

Figure 2: `ParallelLinear` allows for performing different combinations of SMoE transformations allowing for the input and output to be either grouped or scattered. This basic functionality forms the basis of both forward and backward passes of ScatterMoE. Unlike existing implementations, these operations are performed without additional copying (or padding) of the input and output tensors.

**Other applications of SMoEs**   Aside from MLPs, SMoE versions of the attention module have also been proposed (Zhang et al., 2022; Csordás et al., 2023). These mixture-of-attention (MoA) implementations have been used to scale up Universal Transformers (Dehghani et al., 2018; Tan et al., 2023), and also for applications to continual learning in a fully modularised Transformer (Shen et al., 2023). Computing SMoEs efficiently will bring huge benefits to the training and inference of these models.

## 3   Method

In this paper, we will maintain the notation convention of boldface for matrices, *i.e.* $\mathbf{X}$. Unless otherwise stated, the first dimension is batch-time (batch and time dimensions flattened) for ease of understanding. Additionally, $\mathbf{X}_i$ denotes the $i$-th row of $\mathbf{X}$.

### 3.1   Sparse Mixture-of-Experts

SMoE modules are made up of *E experts* which are typically sub-modules of a similar architecture. Each of the $T$ tokens in the input is routed via a routing module, and then based on the router output weights, assigned to $k$ experts, where $k \leq E$. However, the naive method of computing an SMoE (iterating over all tokens and evaluating the respective expert output) is far too slow, and does not exploit the full parallelism of GPU computation. In practice, SMoE implementations often perform the following main steps:

1. **Routing** – Based on each token embedding $\mathbf{X}_t$, the router assigns the weights for each expert $g\left(\mathbf{X}_t\right)$, and only the top-$k$ experts are selected.

2. **Grouping** – This step groups all tokens that are assigned to the same expert together. If $k > 1$, as is often the case, then this also results in a "fan out" of tokens, resulting in $kN$ embeddings in total.

3. **Expert transform** – Now that the tokens are grouped by expert, each expert (a linear transform) can be efficiently computed by batched vector transformations (matrix-matrix multiplications).

4. **Scattering** – This step returns each token to be grouped by its original time-step. This still results in a $kN$ embeddings in total.

5. **Weighted sum** – This step combines the $k$ outputs per token by its original routing weight,

$$\mathbf{Y}_t = \sum_{e \in \text{topk}(g(\mathbf{X}_t))} g_e(\mathbf{X}_t) \cdot f_e(\mathbf{X}_t)$$

resulting in $N$ embeddings.

Typically, each expert is a Multi-layer Perceptron (MLP) with one hidden layer of dimension $d_{\text{expert}}$.

In Megablocks, steps (1) and (4) result in a copy of the original input. They further pad the per-expert blocks of tokens so that they fit into equal count for convenient GPU computation (See Figure 1). This allocates the padded array for the embeddings in High Bandwidth Memory (HBM) and copies the original embeddings in sorted order into it. A Grouped GeMM is then performed on the expert-sorted array.

ScatterMoE, on the other hand, avoids realising the entire padded array in HBM. Instead of copying all the embeddings into a padded array, we sort the tokens according to the experts, and pad the indices instead. When loading a tile into Static RAM (SRAM), we load according to the padded indices, resulting in a padded tile.

## 3.2 `ParallelLinear` **operation**

Our implementation of SMoE relies on `ParallelLinear`, which allows for different combinations of *grouped* General Matrix Multiplications (GeMMs). In order to achieve this, we wrote a Triton kernel, `scatter2scatter`, that enables all combinations of operations shown in Figure 2. This operation fuses grouped GeMMs and scattered read and write operations, which allows us to skip an intermediate group and copy step. `ParallelLinear` allows options for grouped and scattered for both input and output, resulting in the four possible combinations seein in Figure 2. With combinations of these operations, we can implement both the forward and backward passes of `ParallelLinear`.

Algorithm 1 provides the pseudo-code of `ParallelLinear`. It is a thin wrapper around the scatter2scatter kernel, and the grouping options are provided as arguments to `ParallelLinear`. This is the main workhorse of our SMoE implementation, and can be used for both implementing an MLP and an attention layer. We implemented the backward pass of `ParallelLinear` independently of its downstream usage to allow it to be used as a primitive to build other experts upon.

---

**Algorithm 1** ParallelLinear FORWARD

---

**Input:**

| | | | | | |
|---|---|---|---|---|---|
| $\mathbf{X}$ | $T \times d_{\text{in}}$ | input matrix | $\mathbf{o}$ | $T$ | order indices |
| $\mathbf{W}$ | $E \times d_{\text{in}} \times d_{\text{out}}$ | transform tensor | $k$ | | top-$k$ |
| | | | | | *default $k = 1$* |
| $\mathbf{p}$ | $S \times j$ | routing weights | | | |
| | where $Sj = Tk$ | *default $\mathbf{p} : (Tk \times 1) = \mathbf{1}$* | | | |
| *options | | | | | |
| | grouped_in | True, False | grouped_out | True, False | |

**Output:**

| | | |
|---|---|---|
| $\mathbf{Y}$ | $S \times d_{\text{out}}$ | output matrix |

$\hat{\mathbf{Y}} \leftarrow \text{scatter2scatter}(\mathbf{X}, \mathbf{W}, \mathbf{o}, k, *\text{options})$
**if** $\mathbf{p} \neq \varnothing$ **then**
    $\hat{\mathbf{Y}} \leftarrow \text{view}(\hat{\mathbf{Y}}, S, j, -1)$                 *// reshape and weighted sum if $\mathbf{p}$ is provided.*
    $\mathbf{Y} \leftarrow \text{bmm}(\mathbf{p}, \hat{\mathbf{Y}})$
**else**
    $\mathbf{Y} \leftarrow \hat{\mathbf{Y}}$
**end if**

---

### 3.2.1 Backward pass

In a typical batched linear transformation $\mathbf{Y} = \mathbf{XW}$, we need to compute the gradients $\nabla \mathbf{X}$ and $\nabla \mathbf{W}$.

$$\nabla \mathbf{X} = \nabla \mathbf{Y} \mathbf{W}^\top, \qquad\qquad \nabla \mathbf{W} = \mathbf{X}^\top \nabla \mathbf{Y},$$

In `ParallelLinear`, we will need to compute these gradients for each of the $E$ experts. While this could be computed in a way where $\mathbf{X}$ and $\nabla \mathbf{Y}$ are both scattered, the implementation of this operation is fastest when both $\mathbf{X}$ and $\nabla \mathbf{Y}$ are grouped[5] .

---

**Algorithm 2** `ParallelLinear` BACKWARD

---

**Input:**

| | | | | | | |
|---|---|---|---|---|---|---|
| $\nabla \mathbf{Y}$ | $S \times d_{\text{out}}$ | gradient matrix | $\hat{\mathbf{Y}}$ | $Tk \times d_{\text{out}}$ | from forward pass |
| $\mathbf{W}$ | $E \times d_{\text{in}} \times d_{\text{out}}$ | transform tensor | $\mathbf{X}$ | $T \times d_{\text{in}}$ | from forward pass |
| $k$ | | top-$k$ | $\mathbf{o}$ | $Tk$ | from forward pass |
| | | *default $k = 1$* | | | |
| column $\mathbf{p}$ | $S \times j$ | routing weights | | | |
| | where $Sj = Tk$ | *default $\mathbf{p} : (T \times 1) = \mathbf{1}$* | | | |

**Output:**

| | | | | |
|---|---|---|---|---|
| $\nabla \mathbf{X}$ | | same size as $\mathbf{X}$ | $\nabla \mathbf{W}$ | same size as $\mathbf{W}$ |
| $\nabla \mathbf{p}$ | | same size as $\mathbf{p}$ | | |

**if** $\mathbf{p} \neq \varnothing$ **then**
  $\nabla \mathbf{p} \leftarrow \text{bmm}\left(\nabla \mathbf{Y}, \hat{\mathbf{Y}}\right)$
  $\bar{\nabla} \mathbf{Y} \leftarrow \text{group}(\nabla \mathbf{Y}, \mathbf{o}, \mathbf{p})$                       *// weight and group*
**else**
  $\bar{\nabla} \mathbf{Y} \leftarrow \mathbf{Y}$
**end if**
**if** $\mathbf{X}$ is not grouped **then**
  $\bar{\mathbf{X}} \leftarrow \text{group}(\mathbf{X}, \mathbf{o}, \varnothing)$
**else**
  $\bar{\mathbf{X}} \leftarrow \mathbf{X}$
**end if**
$\nabla \mathbf{W} \leftarrow \text{groupXTY}\left(\bar{\mathbf{X}}, \bar{\nabla} \mathbf{Y}\right)$
$\bar{\nabla} \mathbf{X} \leftarrow \text{scatter2scatter}\left(\bar{\nabla} \mathbf{Y}, \mathbf{W}^\top, \mathbf{o}, 1\right)$      *// grouped to scatter or group depending on original input*

---

Algorithm 2 groups the embeddings when they are not grouped in order to compute $\nabla \mathbf{W}$. `groupXTY` is the kernel that implements this grouped matrix multiplication. While this grouping incurs additional memory allocations for potentially very large matrices, these array allocations could be reused. Once the gradients $\nabla \mathbf{p}$ has been computed, the array for $\hat{\mathbf{Y}}$ can be used as the output for $\bar{\nabla} \mathbf{Y}$. $\bar{\mathbf{X}}$ can be reused for $\bar{\nabla} \mathbf{X}$ as they are of the same dimensions. We can then further minimise the use of memory during the backward pass by re-using the arrays used for the grouping operations. We colour the reused arrays in blue and orange respectively in Algorithm 2.

### 3.2.2 SMoE Multi-layer Perceptron (SMoE MLP)

In the context of an SMoE MLP, we can reduce the memory footprint even further. The MLP requires two linear transformations, and could be naively implemented with two `ParallelLinear` operations set to perform scatter-to-scatter transformations. However, we can configure these two linear transforms to be scattered-to-grouped then grouped-to-scattered respectively. This will allow each `ParallelLinear` transform in the SMoE MLP to require only one group operation during the backward pass.

### 3.3 Extensibility: Mixture-of-Attention (MoA)

There have been several proposals for applying SMoEs to the attention layer (Zhang et al., 2022; Csordás et al., 2023; Tan et al., 2023). Regardless of formulation, before the attention

---

[5]We tested `scatterXTY` and found it to be slower than a grouping operation followed by `groupXTY`

---

**Algorithm 3** SMoE MULTI-LAYER PERCEPTRON

---

   **Input:**
     $\mathbf{X}$   $T \times d_{\text{model}}$         input matrix         $\mathbf{o}$   $T$    vector of grouped ordering
     $\mathbf{W}_1$   $E \times d_{\text{model}} \times d_{\text{expert}}$   transformation tensor
     $\mathbf{W}_2$   $E \times d_{\text{expert}} \times d_{\text{model}}$   transformation tensor
   **Output:**
     $\mathbf{Y}$   $T \times d_{\text{model}}$         output matrix
   $\mathbf{H} \leftarrow \texttt{ParallelLinear}\left(\mathbf{X}, \mathbf{W}_1, \mathbf{o}, \varnothing, \texttt{grouped\_in=False}, \texttt{grouped\_out=True}\right)$
   $\mathbf{H} \leftarrow \sigma\left(\hat{\mathbf{H}}\right)$                                                                   *// where $\sigma$ is any point-wise non-linearity*
   $\mathbf{Y} \leftarrow \texttt{ParallelLinear}\left(\mathbf{H}, \mathbf{W}_2, \mathbf{o}, \mathbf{p}, \texttt{grouped\_in=True}, \texttt{grouped\_out=False}\right)$

---

layer is applied, the embeddings should be in chronological order (scattered) to facilitate the application of positional embeddings and to compute the result of the attention weights and value embeddings. With existing SMoE implementations, there would be an additional pair of group-scatter operations, incurring additional memory costs.

ScatterMoE provides an advantage. Since we can retain the scattered ordering through a `ParallelLinear` transform, we can implement MoAs without allocating the extra arrays for grouping and scattering. Figure 3 shows the operations used for SMoE Attention. In this report, we specifically implement and benchmark the Mixture of Multi-head Attention variant found in Tan et al. (2023). This version resembles Grouped-query Attention (GQA; Ainslie et al. 2023), where each key head has multiple possible query heads, while in the SMoE setting, there would be $h_{\text{expert}}$ key heads, with $k \cdot h_{\text{expert}}$ query heads selected from a possible $E \cdot h_{\text{expert}}$ heads, where $h_{\text{expert}}$ is the number of heads per expert.

Following the standard attention implementation conventions, we set $d_{\text{out}} = d_{\text{expert}} \cdot d_{\text{head}}$, where $d_{\text{head}}$ is the number of dimensions for each head. We then reshape accordingly when we perform the attention operation so that the separate heads interact independently.

In Algorithm 4, we also consider the time dimension, so we express the inputs and intermediate arrays as 3-dimensional tensors with the dimensions for batch, time, and embedding dimensions ($B \times T \times d$). In practice, we assume that the input is contiguous and is batch-time ordered, allowing us to flatten the tensor and proceed as

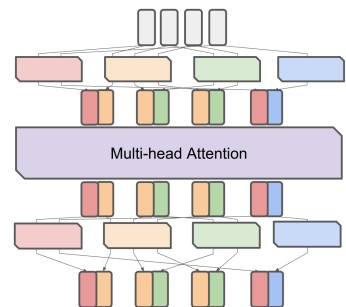

Figure 3: `ParallelLinear` allows for scattered to scattered transformations which retains the chronological order.

we did in the case of the MLP. Note that for the SMoE attention, a key distinction is that we require `ParallelLinear` to give an ungrouped output after the first transformation, and it takes an ungrouped input for the output transform, which means both `ParallelLinear` transformation use a scattered to scattered configuration (Figure 2c).

## 4 Performance

In this section, we cover the performance of our implementation for both training and inference. As an overall integrated test, we benchmark our method within Mixtral (Jiang et al., 2024), using a ~1.5B parameter configuration,

$$d_{\text{model}} = 1024, \qquad d_{\text{expert}} = 3584, \qquad k = 2, \qquad E = 8, \qquad L = 16,$$

We compare against the naive implementation from HuggingFace (Naive HF impl.), then swapping out the SMoE layer with Megablocks sparse (MB (Sparse)) and grouped memory efficient (MB (Mem. eff.)), and finally ScatterMoE (Ours). Our goal is to measure the overall throughput in a training setting.

We ran the training for 100 training updates, with an effective batch size of 256 and 2048 tokens per instance, across 8 A100 GPUs on the same compute node. For both the naive and

---

**Algorithm 4** SMoE MULTI-HEAD ATTENTION

---

**Input:**
  $\mathbf{X}$  $B \times T \times d_{\text{model}}$  input matrix  $k$  top-$k$
  $\mathbf{o}$  $BTk$  grouped indices  $\mathbf{p}$  $B \times T \times k$  router weights
  $\mathbf{W}_K$  $d_{\text{model}} \times d_{\text{out}}$  key transform  $\mathbf{W}_V$  $d_{\text{model}} \times d_{\text{out}}$  value transform
  $\mathbf{W}_Q$  $E \times d_{\text{model}} \times d_{\text{out}}$  query transform  $\mathbf{W}_O$  $E \times d_{\text{out}} \times d_{\text{model}}$  output transform
**Output:**
  $\mathbf{O}$  $B \times T \times d_{\text{model}}$  output matrix

$\mathbf{V} \leftarrow \mathbf{X}^\top \mathbf{W}_V$
$\mathbf{K} \leftarrow \mathbf{X}^\top \mathbf{W}_K$
$\mathbf{Q} \leftarrow \texttt{ParallelLinear}\left(\mathbf{X}, \mathbf{W}_Q, \mathbf{o}, \varnothing, \texttt{grouped\_in=False}, \texttt{grouped\_out=False}\right)$
$\hat{\mathbf{O}} \leftarrow \texttt{Attention}\left(\mathbf{Q}, \mathbf{K}, \mathbf{V}\right)$
$\mathbf{O} \leftarrow \texttt{ParallelLinear}\left(\hat{\mathbf{O}}, \mathbf{W}_O, \mathbf{o}, \mathbf{p}, \texttt{grouped\_in=False}, \texttt{grouped\_out=False}\right)$

---

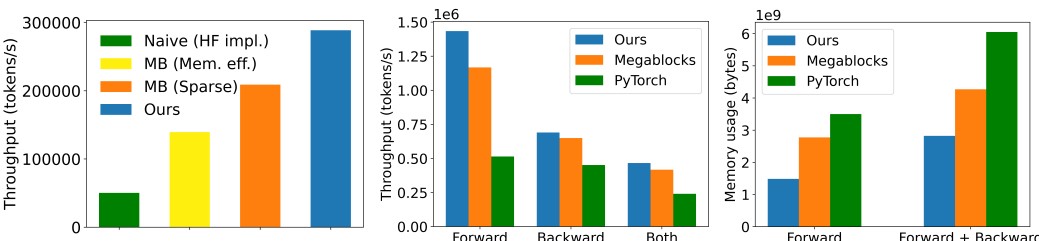

(a) 1.5B model training throughput (b) SMoE MLP unit throughput (c) SMoE MLP unit memory use

Figure 4

ScatterMoE, this resulted in an actual batch size of 128 and 2 accumulation steps, while the Megablocks benchmarks required a batch size if 64 and 4 accumulation steps. We ran the training for a 100 steps and computed the overall throughput. Our method outperforms both the Sparse Megablocks implementation by **38.1%** in this setting. This indicates the importance of the smaller memory footprint, but at larger dimensions with equivalent batch sizes, the gains are not as significant.

The rest of this section covers the other benchmarking experiments we performed in more detail, testing for the effects of decreasing sparsity, and increasing granularity of experts, and benchmarking of our Mixture of Attention implementation.

### 4.1 Unit Benchmarking on the SMoE MLP

Unless otherwise stated, we use the following model hyperparameters,

$$d_{\text{model}} = 4096, \qquad d_{\text{ff}} = 2d_{\text{model}}, \qquad d_{\text{expert}} = d_{\text{ff}}/k, \qquad E = 8k$$

For example, the active hidden units for an MLP is $2 \cdot 4096 = 8192$. If $k = 4$, then $E = 4 \cdot 8 = 32$, with each expert being $8192/k = 2048$ dimensions. Each datapoint on the plot is the median and 5-th and 95-th percentiles of 100 runs of the module. In this unit test, we use a more efficient PyTorch implementation from the implementation of Tan et al. (2023) [6]

Figure 4a summarises the overall performance for an SMoE MLP where $E = 32, k = 4$, and $T = 30 \cdot 2048$. All benchmark times are measured on an Nvidia A100 GPU. Generally, we find that ScatterMoE has slightly higher throughput during training, for the same input sizes. Our method shows a larger margin of improvement for inference.

On memory consumption, our implementation for the SMoE MLP uses **66.2%** of the memory Megablocks uses during training, while using only **53.6%** of the memory of Megablocks if we consider only inference.

---

[6] https://github.com/shawntan/SUT/tree/main/sut_layer/parallel_linear/parallel_experts

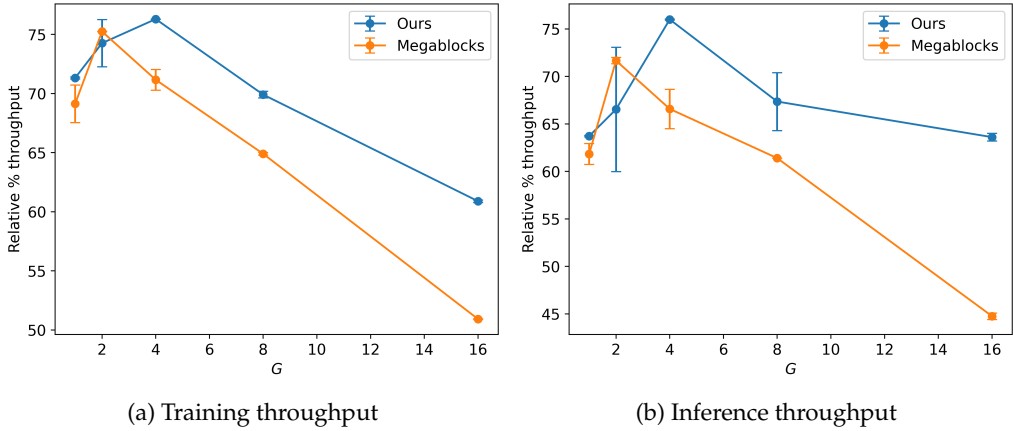

(a) Training throughput            (b) Inference throughput

Figure 5: Increasing $k$ and $E$ while fixing the number of active parameters and total parameters. We find that our implementation scales better with higher $k$. Inference granularity scaling performance. The difference in relative throughput is higher if we consider only the forward pass.

## 4.2 Granularity and Throughput

Krajewski et al. (2024) defines the concept of granularity. Given a SMoE MLP with the equivalent active parameters as an MLP with intermediate dimension layer of dimension $d_{\text{ff}}$, and with each expert of dimension $d_{\text{expert}}$, then granularity is defined as,

$$G = \frac{d_{\text{ff}}}{d_{\text{expert}}}$$

Here, we measure the effect of throughput as we vary $G$ while fixing $d_{\text{ff}}$. Accordingly, with higher values of $G$, we need to increase values of $k$ and $E$ — more granularity requires more experts to achieve the same active parameters.

We test values of $k \in \{1, 2, 4, 8, 16\}$, and $E = 8k$ for divisibility of dimension sizes. Figure 5 shows how throughput of both methods vary relative to a model with equivalent active parameters. Since we maintain constant active and total parameters for these runs, these results are also constant for all $G$.

We find that ScatterMoE scales with $G$ with better throughput. This may be related to the increase in zero-padding required for higher number of $E$ in the case of Megablocks — as the number of experts grows and the embeddings assigned to each expert decreases, there will be more padding introduced. If we consider only the forward pass, the relative gap between our method and Megablocks is also much higher than in the case of training. This makes our method favourable for batched inference, especially with high granularity settings. The results of Krajewski et al. (2024) suggests higher $G$ for SMoE models, and our implementation seems suited for this application.

## 4.3 Decreasing sparsity

We can view the SMoE as an interpolation between a model with the size of just the active parameters $k \cdot d_{\text{expert}}$ and a large fully dense model with $E \cdot d_{\text{expert}}$. However, SMoE comes with additional overhead (*e.g.* routing, sorting), and we want to measure how much reducing sparsity will affect throughput, in comparison to a fully dense model.

In this experiment, we tested on increasing values of $k \leq 30$. Further increasing $k$ reaches the limit of device memory for Megablocks. We maintain $E = 64$ for all runs, and compare the performance of both Megablocks and ScatterMoE to a dense model with a $d_{\text{ff}} = E \cdot d_{\text{expert}}$.

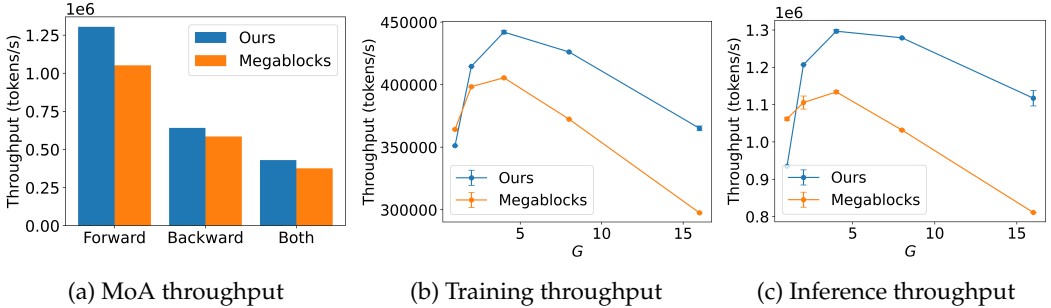

(a) MoA throughput    (b) Training throughput    (c) Inference throughput

Figure 8: The curves for increasing granularity for the MoMHA implementation. In this case, the Active params baseline varies in throughput because of the shared key and value vectors across the experts.

Generally, we find that while our implementation performs with slightly better throughput overall, both Megablocks and our implementations are still more efficient than a large dense MLP with equivalent parameters. However, note that in this case, the throughput for $k = 30, E = 64$ is already reaching the throughput for an equivalent dense model with the same parameters. Further increasing $k$ exceeds the memory of the device we ran the benchmarks on for Megablocks.

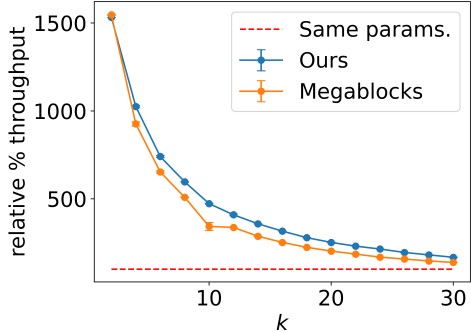

Figure 6: Relative throughput curves as we decrease sparsity (increasing $k$)

## 4.4 Mixture-of-Attention

As previously mentioned, we implement the Mixture of Multi-head Attention (MoMHA) variant of Attention SMoEs as described in Tan et al. (2023). This implementation allows for multiple attention heads per expert, and shares the key and value embeddings across the attention experts. This formulation is similar to Grouped-query Attention (GQA; Ainslie et al. 2023), where each head of the key has several query heads, each of these forming a group. MoMHA experts are the equivalent of groups in GQA, where each group is of size $k$.

For the following benchmarks, we adhere to the following parameters:

$$d_{\text{model}} = 4096, \quad d_{\text{head}} = 128, \quad T = 16 \cdot 2048, \quad h = 32, \quad h_{\text{expert}} = h/k, \quad E = 8k$$

where $d_{\text{head}}$ is the dimension of each attention head and $h$ is the number of *active* attention heads.

We implemented an equivalent baseline in Megablocks using the 'dense' configuration in the library. This version still suffers from the issue with having to perform redundant grouping and scattering steps.

For $k = 8$, we find that our implementation outperforms Megablocks, by **24.0%** of throughput for inference. We also note from Figure 8, that as we increase granularity (fewer heads per expert / smaller $h_{\text{expert}}$), the gap between our implementation and Megablocks grows as well. Again, our method is favourable for use in high granularity settings for Mixture-of-Attention

| Tasks | Metric | Hugging Face | ScatterMoE | Abs. Error |
|---|---|---|---|---|
| winogrande | | 0.7632 | 0.7640 | 0.0008 |
| sciq | | 0.9520 | 0.9580 | 0.0060 |
| race | | 0.4057 | 0.4010 | 0.0047 |
| piqa | | 0.8330 | 0.8368 | 0.0038 |
| openbookqa | accuracy | 0.4680 | 0.4740 | 0.0060 |
| hellaswag | | 0.8396 | 0.8405 | 0.0009 |
| copa | | 0.9300 | 0.9300 | 0.0000 |
| boolq | | 0.8523 | 0.8541 | 0.0018 |
| arc_easy | | 0.8350 | 0.8350 | 0.0000 |
| arc_challenge | | 0.5973 | 0.5981 | 0.0008 |
| wikitext | perplexity | 5.6135 | 5.6142 | 0.0007 |

Table 1: Language Model Evaluation Harness results comparisons: Hugging Face v. Scatter-MoE implementations. Differences in results between both implementations are negligible.

## 4.5 Mixtral Inference Comparison

Finally, we converted Mixtral 8x7B[7] to use ScatterMoE, and ran the LM Evaluation Harness (Gao et al., 2023) on several benchmarks (See Table 1). As ScatterMoE is an alternative implementation of Sparse MoEs, we do not expect any differences in the final evaluation results. The absolute error between the Hugging Face naive implementation and ScatterMoE is sufficiently small, which demonstrates this.

We have included the script to convert the model parameters from Hugging Face's format to a format compatible for ScatterMoE[8]

## 5 Conclusion & Limitations

ScatterMoE is an implementation of SMoEs in Triton that reduces the memory footprint and offers slightly higher throughput on the GPU compared to existing solutions. We have also engineered ScatterMoE to use the `ParallelLinear` primitive, which we envision to be a module that can be extended upon to build other SMoE-style modules that require grouped or scattered linear transformations.

At present, ScatterMoE does not implement a specialised kernel for speeding up decoding, and further work is required for parallelisation in a multi-node setting. These additional features will be added in future iterations, and we believe further testing by us and the open source community will iron out any remaining bugs and performance issues left unoptimised.

Finally, we have also provided an implementation of an MLP and Attention layer based on ScatterMoE, that we hope will benefit any future implementations of Mixture-of-Expert based models, and serve as worked examples for extending the concept of SMoEs to other variants of linear transformation based experts.

### Acknowledgments

We would like to Bowen Pan and Mayank Mishra for their testing and feedback of Scatter-MoE. Songlin Yang also gave valuable advice during the development of the ScatterMoE kernels.

---

[7]https://huggingface.co/mistralai/Mixtral-8x7B-v0.1
[8]https://github.com/shawntan/scattermoe/blob/main/examples/convert.py

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
