# OpenReview forum: "Scattered Mixture-of-Experts Implementation"
_colmweb.org/COLM/2024/Conference — COLM_

### Official Review · Reviewer_dRgj · 2024-05-08

**Rating:** 8
**Confidence:** 3
**Ethics Flag:** 1

**Summary:**

This paper presents ScatterMoE, an SMoE implementation that minimizes memory overhead. Since ParallelLinear allows for performing different combinations of SMoE transformations allowing for the input and output to be either grouped or scattered, ScatterMoE relies on ParallelLinear to achieve both forward and backward passes. Generally, these operations are performed without additional copying (or padding) of the input and output tensors involved in the existing implementations. Experiments validate the effectiveness of ScatterMoE on throughput and memory use.

**Questions To Authors:**

1) For Megablocks and PIT, the authors stated that “the sparse matrix format makes the intermediate representation harder to extend upon”. Does this lead to memory overhead? If yes, can you describe the reasons? Otherwise, is there any in-depth analysis or experiments on the aforementioned feature of Megablocks and PIT?

2) In the comparative experiments, what is the performance of PIT?

**Reasons To Accept:**

1) The implementation proposed in this study is beneficial for various real-world applications.

2) The paper is written-well with a good presentation style and experimental results.

**Reasons To Reject:**

I have several questions to authors, as listed in the following part.

---

> ### Author Rebuttal · Authors · 2024-05-30
>
> > For Megablocks and PIT, the authors stated that “the sparse matrix format makes the intermediate representation harder to extend upon”. Does this lead to memory overhead? If yes, can you describe the reasons? Otherwise, is there any in-depth analysis or experiments on the aforementioned feature of Megablocks and PIT?
>
> In an MoE MLP, the sparse matrix format does not create the overhead.  Megablocks makes a re-ordered copy of the input before performing the GeMM operation, which is the step that creates the additional memory overhead.  Figure 1 in the paper illustrates the additional copying that causes this.
>
> However, extending MoE to Attention the memory issue is further compounded:
>
> 1. First we need to perform a ParallelLinear transformation of the input, to produce queries, which will require re-ordering to *expert order.*
> 2. These queries will need to be re-ordered to *chronological order* for application of position embeddings (e.g. RoPE) and appropriate masking.
> 3. Once the standard attention step is done, the output embeddings will have to be reordered to *expert order* again,
> 4. before the final output expert transformation, before being reordered to *chronological order*.
>
> There are 4 re-ordering steps in total, and each will incur an additional duplication of the input. With scatter2scatter, we can bypass the reordering by only performing the expert ordered operation in SRAM. This results in a 42% savings in memory usage.
>
> Both Megablocks (in sparse mode) and PIT rely on a sparse representation of the intermediate hidden state in an MoE MLP. Unless the operation performed on the sparse representation is a point-wise operation (like an activation function, for example) to perform anything that requires order sensitivity (attention) would require converting back to the dense format, then re-ordering, incurring all of the above-mentioned costs AND the sparse-dense-sparse conversion.
>
> > In the comparative experiments, what is the performance of PIT?
>
> We were unable to find an open source implementation of PIT to compare against

---

> > ### Comment · Reviewer_dRgj · 2024-06-05
> >
> > Thanks for your responses. I intend to keep my original score.

---

> > > ### Author Response · Authors · 2024-06-06
> > > **Thank you!**
> > >
> > > We appreciate your feedback and it will help us to improve our paper. Thank you!

---

### Official Review · Reviewer_4BFX · 2024-05-10

**Rating:** 6
**Confidence:** 4
**Ethics Flag:** 1

**Summary:**

This paper introduces a new method called ScatterMoE for the efficient implementation of MoE. It proposes a new kernel called scatter2scatter which supports group-to-group, group-to-scatter, scatter-to-scatter, and scatter-to-group operation. The benchmark against the recent SOTA work MegaBlocks demonstrates its higher throughput and lower memory footprint.

**Reasons To Accept:**

1. From the interpretation in the paper, this method is simple by fusing fuses grouped GeMMs and scattered read and write operations and effective by higher throughput and lower memory footprint.
2. The paper is clearly written and easy to read.

**Reasons To Reject:**

1. The paper should provide more information about the details of the key innovation, including its detailed implementation, how it avoids padding, instead of using paragraphs to introduce MoE, its integration with MoE and MoA. Also, the pseudo-codes are not informative enough.
2. It is advised to provide more experimental results about model quality to ensure that using ScatterMoE can achieve similar or even better quality on the LLM evaluation benchmarks.

---

> ### Author Rebuttal · Authors · 2024-05-30
>
> Thank you for the feedback!
> > The paper should provide more information about the details of the key innovation, including its detailed implementation, how it avoids padding, instead of using paragraphs to introduce MoE, its integration with MoE and MoA. Also, the pseudo-codes are not informative enough.
>
> We will include more detailed information about the specifics you pointed out. Specifically, we will expand on the following points in the paper:
> 1. Instead of performing a padded copy in High Bandwidth Memory (HBM), we sort the tokens according to the experts, and pad the indices instead.
> 2. When loading a tile into Static RAM (SRAM), we load according to the padded indices, resulting in a padded tile. The full padded array of token embeddings is never realised in HBM.
> 3. We will add the pseudo-code for the scatter2scatter kernel itself, since the current pseudo-code only describes the overarching process using scatter2scatter as a primitive.
>
> > It is advised to provide more experimental results about model quality to ensure that using ScatterMoE can achieve similar or even better quality on the LLM evaluation benchmarks.
>
> We converted Mixtral 7B to use ScatterMoE (script included), and ran the lm-evaluation harness on the following benchmarks.
>
> |     Tasks     |      Metric     |   HF   | Scattermoe |
> |:-------------:|:---------------:|:------:|-----------:|
> | winogrande    | acc             | 0.7632 |     0.7640 |
> | wikitext      | word_perplexity | 5.6135 |     5.6142 |
> | sciq          | acc_norm        | 0.9520 |     0.9580 |
> | race          | acc             | 0.4057 |     0.4010 |
> | piqa          | acc_norm        | 0.8330 |     0.8368 |
> | openbookqa    | acc_norm        | 0.4680 |     0.4740 |
> | hellaswag     | acc_norm        | 0.8396 |     0.8405 |
> | copa          | acc             | 0.9300 |     0.9300 |
> | boolq         | acc             | 0.8523 |     0.8541 |
> | arc_easy      | acc_norm        | 0.8350 |     0.8350 |
> | arc_challenge | acc_norm        | 0.5973 |     0.5981 |
>
> The difference in the results are negligible, which we expect as ScatterMoE is a different implementation and does not make any fundamental changes to MoEs.

---

### Official Review · Reviewer_9gxi · 2024-05-12

**Rating:** 6
**Confidence:** 4
**Ethics Flag:** 1

**Summary:**

This paper proposes ScatterMoE which optimizes the memory cost and the throughput of tokens for SMoE models. Specially, ScatterMoE focuses on the issue on extra memory copies of token representations during the operation of MoE scattering and gathering. By implementing custom kernels using Triton or CUTLASS, ScatterMoE eliminates the extra memory copies and improves the overall training & inference performance, compared with existing SMoE frameworks like Megablocks. The work provides the detailed implementation of ScatterMoE and extensively compares the performance with baselines (Megablocks and naive PyTorch implementation) considering a variety of experimental settings (granularity, sparsity and MoA).

**Questions To Authors:**

For some recent variants of MoE like shared experts (DeepSeekMoE for e.g.), will ScatterMoE still be supported and in advantage in performance?

**Reasons To Accept:**

1. The motivation is straightforward and reasonable, which directly eliminates the excessive memory consumption of representation tensor copies and token paddings.
2. The paper describes the implementation clearly including the forward & backward passes of ParallelLinear and its extension to MoA modules.
3. The comparison in performance provided is extensive and includes sufficient conditions (like granularity and sparsity) which affects the experimental results.

**Reasons To Reject:**

1. As a competitor of Megablocks, this work should better be accomplished into a usable package (instead of just raw fused kernels) and provide a more end-to-end time-basis training curve for current prevalent LLMs. If achieved, I think the work will be much more influential and have more significance to be accepted.
2. It will be much better if larger scales of MoE models are benchmarked, in fact a 1.5B parameter configuration is more like a toy setting.

---

> ### Author Rebuttal · Authors · 2024-05-30
>
> Thank you for your feedback!
>
> > As a competitor of Megablocks,... , I think the work will be much more influential and have more significance to be accepted.
>
> We agree with this perspective,  and ScatterMoE has been integrated into an open source fine-tuning framework and we have plans to include it into a pretraining framework are underway (pull-request submitted).
>
> However,  ScatterMoE is a modular, light-weight implementation of MoEs that is meant to  be used as a drop-in replacement for MoEs. We’ve added a conversion script for the Mixtral model on huggingface to demonstrate this, and include the lm-evaluation-harness results achieved with both the naive huggingface implementation and using ScatterMoE, and the results show that the differences are negligible.
>
> |     Tasks     |      Metric     |   HF   | Scattermoe |
> |:-------------:|:---------------:|:------:|-----------:|
> | winogrande    | acc             | 0.7632 |     0.7640 |
> | wikitext      | word_perplexity | 5.6135 |     5.6142 |
> | sciq          | acc_norm        | 0.9520 |     0.9580 |
> | race          | acc             | 0.4057 |     0.4010 |
> | piqa          | acc_norm        | 0.8330 |     0.8368 |
> | openbookqa    | acc_norm        | 0.4680 |     0.4740 |
> | hellaswag     | acc_norm        | 0.8396 |     0.8405 |
> | copa          | acc             | 0.9300 |     0.9300 |
> | boolq         | acc             | 0.8523 |     0.8541 |
> | arc_easy      | acc_norm        | 0.8350 |     0.8350 |
> | arc_challenge | acc_norm        | 0.5973 |     0.5981 |
>
> > It will be much better if larger scales of MoE models are benchmarked....
>
> We benchmarked a bigger model of 9B parameters: 32 experts and k=4, d_model = 2048, mlp_size = 1536 per expert, 26 layers. This was the largest we could manage with a single node and 8 X A100 80gb GPUs. At this size, the throughput for a minibatch of 32 * 2048, with 16 acc steps for an actual batch size of 2, was 57957.8 tokens/s for ScatterMoE, and 19924.9 tokens/s for Megablocks scattered. However, this may be due to a memory leak in the recent version of Megablocks.
>
> > For some recent variants of MoE like shared experts (DeepSeekMoE for e.g.),...
>
> ScatterMoE can support the shared expert scheme in DeepSeekMoE: one simple way to do this is to either adjust the logit to a large value for that expert, or to append the shared expert index before passing it to the MoE module.  However, it is unlikely that there will be an advantage over those already discussed in this report.

---

### Decision · Program_Chairs · 2024-07-10

**Decision:**

Accept

**Comment:**

This work proposes a new approach for a more efficient implementation of sparse MoE models by reducing memory copies, particularly compared with existing frameworks such as Megablocks.

Pros: The approach is straightforward and essentially works by removing extraneous memory copies. The approach is cleanly described and there are thorough performance comparisons with alternative approaches such as Megablocks. The approach performs well even up to 9B model size.

Cons: It would be great to have this integrated into a useable pretraining package.

Overall, I think this would be a great contribution to the conference.